# Research on deep learning-based water quality prediction and management for intelligent aquaculture systems

Anonymous Full Paper
Submission 42

## Abstract

This study develops a deep learning model for predicting nitrogen compound concentrations in aquaculture systems using time-series data. The LSTM-based approach significantly improves water quality prediction accuracy and enables intelligent management for sustainable fish farming.

## 1 Introduction

The aquaculture industry is expected to account for 57% of global fisheries production by 2023, establishing itself as a sustainable food source [1, 2]. In particular, as the instability of fishery resources intensifies due to environmental pollution and climate change, aquaculture is gaining attention as a key industry for future food security. However, securing a healthy growth environment for fish continues to pose a significant challenge, especially under high-density farming conditions, limited water resources, and closed tank systems.

Among them, nitrogen compounds such as ammonia ($NH_3$), nitrite ($NO_2^-$), and nitrate ($NO_3^-$) are the main toxic elements in aquaculture tanks, which inhibit the physiological functions of fish, causing increased mortality and growth decline. Levit [3] synthesized numerous literatures and pointed out that ammonia easily penetrates the tissues through fish gills and can have fatal effects even at extremely low concentrations. Kır et al. [4] reported in an experiment with European sea bass (Dicentrarchus labrax) that increased ammonia concentration and salinity changes had a complex effect on the metabolic rate of fish and induced acute toxicity responses.

In the case of nitrite, Gao et al. [5] reported that nitrite stress affected growth rate, liver metabolism, and detoxification pathways in striped bass (Lateolabrax maculatus) under high temperature conditions, and disrupted the overall physiological metabolic system. Kim et al. [6] also reported that nitrite exposure significantly changed hematological parameters, antioxidant enzyme activities, and stress responses in an experiment using hybrid grouper juvenile individuals.

Nitrate is also attracting attention as an environmental toxicant that can induce chronic toxic reactions. Yu et al. [7] reported that nitrate induces hematological function deterioration, oxidative stress, and apoptosis through long-term exposure experiments on turbot (Scophthalmus maximus). Gao et al. [8] observed similar results through acute exposure experiments on mullet (Takifugu rubripes), reaffirming that nitrate is a potent toxicant that causes loss of ion homeostasis and tissue damage. Water quality prediction research has expanded from statistical models to machine learning and deep learning-based models with technological advancement.

Accordingly, this paper analyzes the causes of the decline in prediction performance from various angles and examines the possibility of prediction failure by comparing structural differences, data characteristics, and model utilization methods with previous studies. Through this, it aims to provide empirical basic data for the development of a water quality prediction system that can be applied to actual aquaculture environments.

## 2 Methodology

The data used in this study were collected from a 9m diameter recirculating aquaculture system (RAS) located in Jeju Island, South Korea. Water quality measurements were performed at 5-minute intervals for approximately 2 years from November 2021 to October 2023, and the main items included water temperature, dissolved oxygen, salinity, pH, oxidation-reduction potential (ORP), suspended solids, nitrate, and total ammonia.

Among these, nitrate ($NO_3^-$) and ammonia ($NH_3$) items were collected in real time through sensors, but the sensors were low-cost sensors with relatively low sensitivity and precision among commercial water quality monitoring devices. Therefore, in this study, in order to secure a baseline for prediction accuracy, water quality at the same time was collected and manually analyzed in the laboratory, and the results were used as reliable correct answers (Ground Truth) for model learning and evaluation.

In general, in the field of water quality prediction, in addition to Min-Max normalization, various scaling techniques such as Z-score standardization, RobustScaler, Log transformation, and Quantile transformation are utilized.

For example, Gao et al. (2023) reported that Z-score normalization was applied to an LSTM-based water quality prediction model to preserve

**Table 1.** Aquaculture water quality data collected from sensors.

| Attribute | Attribute information |
|-----------|----------------------|
| Time | Collected time |
| TP | Water temperature |
| DO | Dissolved oxygen concentraition |
| DF | Dissolved matter content |
| PH | Acidity |
| OR | Oxgeb reduction |
| SL | Salinity |
| NH3 | Ammonia |
| NO2 | Nitrite |
| NO3 | Nitrate |

the variability of time series data while securing prediction stability [1]. Meanwhile, Wang et al. (2022) improved prediction accuracy by applying RobustScaler to water quality data containing many sensor outliers to minimize the influence of extreme values [2]. In addition, Yu et al. (2024) proposed that the prediction performance of a GRU-based model was improved by normalizing non-normally distributed water quality data to a uniform distribution using QuantileTransformer [3].

However, in this study, Min-Max normalization was selected among the normalization techniques. This is because this dataset has relatively few sensor outliers, and the range of each variable is clearly defined in advance, so Min-Max normalization is effective, and both LSTM and Ridge models are optimized for processing input values in the range of 0 to 1, which is advantageous for learning convergence, and when variables such as water temperature, pH, and salinity are normalized within the same scale, learning is possible while maintaining the relative influence between variables.

In order to effectively reflect time series information, a sliding window method was adopted. In this study, the window size was set to 7 days, and data from t–7 to t–1 were configured as an input sequence for prediction of time point t. This structure can reflect local patterns within the time series (e.g., water temperature increase → DO decrease → ammonia increase), and many previous water quality prediction studies have reported that similar approaches are effective [4,5].

## 3 Result

This study attempted to empirically compare the difference in prediction performance between an interpretable linear regression model (Ridge) and a nonlinear model (LSTM) capable of learning time series information, considering the structural characteristics of the two models, as shown in Table 3.

According to the experimental results in Table 3, the Ridge regression model showed very large prediction errors in all three variables (ammonia, nitrite, and nitrate). The mean absolute error (MAE) was 4.27 for ammonia, 20.53 for nitrite, and 12.56 for nitrate, and the root mean square error (RMSE) was also high at 4.48, 23.33, and 12.91, respectively.

On the other hand, the LSTM model showed excellent prediction performance overall. MAE was significantly lower than Ridge at 0.79 for ammonia, 5.05 for nitrite, and 0.61 for nitrate, and RMSE also decreased to 1.07, 7.34, and 1.17, respectively.

**Table 2.** Results of Ridge Regression and LSTM models.

| Model | Value | MAE | RMSE |
|-------|-------|-----|------|
| Ridge Regression | $NH_3$ | 4.2717 | 4.4768 |
| | $NO_2^-$ | 20.5323 | 23.3326 |
| | $NO_3^-$ | 12.5572 | 12.9052 |
| LSTM | $NH_3$ | 0.7914 | 1.0719 |
| | $NO_2^-$ | 5.0544 | 7.3394 |
| | $NO_3^-$ | 0.6143 | 1.1681 |

## 4 Conclusion

This study compared and analyzed the predictive performance of nitrogen compounds (ammonia, nitrite, and nitrate) using a Ridge regression model and an LSTM model, utilizing long-term water quality data collected from a recirculating aquaculture system (RAS) on Jeju Island.

Experimental results showed that the Ridge regression model exhibited relatively high errors across all three variables, limiting its effectiveness in water quality prediction. Conversely, the LSTM model improved overall prediction accuracy by incorporating time-series information, demonstrating particularly stable performance in ammonia and nitrate predictions. This demonstrates the potential of time-series-based deep learning models to reflect the dynamic characteristics of complex water quality data.

However, this study's generalizability is limited due to its reliance on specific aquaculture data and limited sensor environment. Future research requires model validation that considers diverse environmental conditions and data characteristics. Furthermore, ensuring the reliability of sensor data, handling outliers, and utilizing multimodal data are expected to further enhance predictive performance and practicality.

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
