# OpenReview forum: "Research on deep learning-based water quality prediction and management for intelligent aquaculture systems"
_NLDL.org/2026/Abstracts_Track — NLDL 2026 Abstracts_

### Official Review · Reviewer_7yWd · 2025-10-26

**Soundness:** 3
**Correctness:** 3
**Rating:** 4
**Confidence:** 4

**Summary:**

This work investigates the use of deep learning for predicting nitrogen compound levels in aquaculture water from time series data. A classical time series approach is compared to a deep learning approach using an LSTM network, with encouraging performance for the deep learning. The abstract fits the scope of the conference well, and I believe it could encourage valuable discussions on an important application area.

**Strengths:**

1. Interesting application with important implications for industry.
2. Comparison of both classical and deep learning approaches.
3. Clearly written and easy to follow

**Weaknesses:**

All of these are minor weakness:

1. LSTM networks can be unstable to train. It would be beneficial to report the average and standard deviation across several runs to give the reader an impression of the stability of the deep learning model.
2. For future work, it would be interesting to see how recent foundation model for time series data handles such a task, for instance the latest Chronos-2 model [1]

[1] Ansari et al., Chronos-2: From Univariate to Universal Forecasting (https://arxiv.org/abs/2510.15821)

---

### Official Review · Reviewer_aKXB · 2025-10-30

**Soundness:** 3
**Correctness:** 4
**Rating:** 4
**Confidence:** 4

**Summary:**

This paper presents a deep learning approach for predicting nitrogen compound concentrations (ammonia, nitrite, and nitrate) in recirculating aquaculture systems (RAS). The authors develop and evaluate a Long Short-Term Memory (LSTM) model trained on 2 years of real-time water quality data collected at 5-minute intervals from a RAS facility in Jeju Island, South Korea. The LSTM’s performance is compared to a Ridge regression baseline, showing significant improvements in MAE and RMSE across all variables. The study aims to support intelligent aquaculture management by enabling proactive monitoring and control of water quality.

**Strengths:**

Real-world relevance and impact:
The work addresses a critical environmental and industrial problem—accurate water quality prediction for sustainable aquaculture—using real long-term data from a working RAS. The potential for reducing fish mortality and enhancing resource efficiency makes it highly practical.

Empirical rigor:
The dataset covers multiple water quality parameters over a long timescale, providing a robust basis for model training and evaluation. The use of lab-validated ground truth ensures that the model’s accuracy is not biased by low-cost sensor noise.

Clear methodology:
The paper systematically details data collection, normalization methods, and time-series processing using a sliding window of 7 days. This improves interpretability and reproducibility.

Strong quantitative results:
The LSTM achieved large error reductions relative to Ridge regression (e.g., MAE for ammonia reduced from 4.27 to 0.79), demonstrating the clear advantage of time-series deep learning for water quality dynamics.

Balanced discussion:
The authors acknowledge limitations—such as dataset specificity and sensor variability—and suggest future extensions involving multimodal data integration and model generalization.

**Weaknesses:**

Limited novelty in model design:
While the study applies LSTM effectively, it does not propose a new architecture or hybrid method. The contribution lies more in application and evaluation than in algorithmic innovation.

Lack of interpretability and feature analysis:
The model’s inner workings (e.g., which parameters most influence nitrogen prediction) are not explored. Explainable AI (XAI) or feature attribution analysis would strengthen the scientific depth.

No ablation or sensitivity tests:
The effects of window size, normalization method, or data imbalance are not analyzed. These could help validate the robustness of the approach.

Geographic and environmental limitation:
Since the model is trained on a single RAS in Jeju, performance may not generalize to other aquaculture systems with different species or climates.

Presentation and structure:
Figures and tables are informative but somewhat minimal. Including visualizations of time-series trends or error distributions would make the findings more intuitive.

---

### Official Review · Reviewer_tS9Z · 2025-11-03

**Soundness:** 3
**Correctness:** 3
**Rating:** 4
**Confidence:** 3

**Summary:**

This is a comparative study where Ridge regression and LSTM models are benchmarked for prediction of nitrogen compounds in RAS. MAE and RMSE values are calcualted as the prediction performance and it has been shown that LSTM is better than Ridge regression.

**Strengths:**

- The study uses a linear and a non-linear method for the experiments. I believe it might be interesting to present how these approaches perform in predicting the related target values in RAS context.
- The problem is well-defined and the scope of the study is clearly stated.
- The results are presented within the scope of the study and conclusions are made accordingly.
- Since this an abstract study, it is good to highlight what the future research directions may be. And this is addresses clearly in the submission.

**Weaknesses:**

- The cross references of the tables are not correct. Should be checked and corrected. Both tables should be cited in the text.
- The main items for collected data (lines 72-75) are not consistent with the contents of Table-1. For example, no oxidation- 073reduction potential (ORP) in Table-1.
- The model training has been explained in a very brief way. Some more details could have been presented regarding this.

---

### Decision · Program_Chairs · 2025-11-05

**Decision:**

Accept

**Comment:**

The abstract is of interest to the community and should be presented at the conference.